# A Novel Gas Sensor for Detecting Pork Freshness Based on PANI/AgNWs/Silk

**DOI:** 10.3390/foods11152372

**Published:** 2022-08-08

**Authors:** Yahui Li, Yanxiao Li, Jiyong Shi, Zhihua Li, Xin Wang, Xuetao Hu, Yunyun Gong, Xiaobo Zou

**Affiliations:** 1School of Food and Biological Engineering, Jiangsu University, Zhenjiang 212013, China; 2Analytical Instrumentation Center, Jiangsu University, Zhenjiang 212013, China; 3School of Food Science and Nutrition, The University of Leeds, Leeds LS2 9JT, UK

**Keywords:** polyaniline, silver nanowires, silk, trimethylamine, gas sensor, pork freshness

## Abstract

A novel, operational, reliable, flexible gas sensor based on silk fibroin fibers (SFFs) as a substrate was proposed for detecting the freshness of pork. Silk is one of the earliest animal fibers utilized by humans, and SFFs exposed many biological micromolecules on the surface. Thus, the gas sensor was fabricated through polyaniline (PANI) and silver nanowires (AgNWs) and deposited on SFFs by in-suit polymerization. With trimethylamine (TMA) as a model gas, the sensing properties of the PANI/AgNWs/silk composites were examined at room temperature, and the linear correlativity was very prominent between these sensing measures and the TMA measures in the range of 3.33 μg/L–1200 μg/L. When the pork sample is detected by the sensor, it can be classified into fresh or stale pork with the total volatile basic nitrogen (TVB-N) as an index. The result indicated that the gas sensor was effective and showed great potential for applications to detect the freshness of pork.

## 1. Introduction

Pork is a major source of animal protein for humans, and it is the top-consumed meat worldwide [1]. In 2021, pork production was 101,481 kilotons, and pork consumption was 100,853 kilotons; however fresh pork is easily perishable during the storage process. Quality deterioration of meat causes changes in attributes such as surface texture, color, pH, tenderness and odor [2,3]. All attributes are regarded as important parameters for assessing pork quality [4]. Changes in pork freshness are mainly caused by biochemical reactions and microbial spoilage [5,6]. In particular, TBV-N, produced by the breakdown of protein, belongs to the biogenic amine, which includes ammonia, TMA and dimethylamine [7]. TBV-N is the most representative indicator for freshness of meat [8]. Meanwhile, with the decrease in freshness, the pork meat produces smelly TMA, which is an indicator for the assessment of meat freshness.

TMA in meat is usually detected by the spectrometry method [9,10,11]. Recently, gas sensors were used to measure the TMA and freshness in pork as a nondestructive, rapid, sensitive and inexpensive method [12,13,14,15,16]. Several materials have been studied for gas sensors, such as intrinsically conducting polymer, semiconducting metal-oxides and their composites [17,18,19,20]. Silk is a typical natural fiber obtained from silkworm *Bombyx mori* and has been used in textile industries for centuries. Silk represents a unique family of structural proteins that are biocompatible, degradable, and mechanically superior [21]. In this research, silk was used for a substrate in PANI composite materials based on environmental stability, biocompatibility, and morphologic flexibility; therefore, fabricating a miniature, short-period and easy-to-operate at room temperature sensor is significant.

Currently, sensors are developing towards microminiaturization as they are very convenient for practical applications. From 2013 to now, colorimetric array sensors [22,23], film sensors [24,25] and polymer sensors were fabricated to monitor meat freshness. Meanwhile, one-dimensional nanofiber sensors, based on PANI and PANI/metal composite materials, have developed for monitoring food freshness [21,26]. Silver nanowires are regarded as an excellent candidate because of high electrical conductivity, a large specific surface area, and low cost (as reported by Padmanaban) [27].

Therefore, this study presented a promising method for conducting a microsensor based on the PANI, silver and silk. The PANI and AgNWs were successfully deposited on SFFs by in-suit polymerization. PANI/AgNWs/silk composites were prepared for TMA detections and the evaluation of the freshness of pork.

## 2. Materials and Methods

### 2.1. Materials and Apparatus

Silver nitrate (AgNO_3_, 99.0%), anhydrous ethylene glycol (EG, 99.8%), ammonium persulphate (APS, 98.0%), concentrated sulfuric acid (H_2_SO_4_, 98.0%), magnesium oxide (MgO, 99.9%), boric acid (99.8%), methyl red, methylene blue, trimethylamine (TMA·H_2_O, 33.0%), ammonium hydroxide (NH_3_·H_2_O, 25–28%), ethanol (C_2_H_6_O, 99.8%) and sodium carbonate (Na_2_CO_3_, 99.8%) were purchased from Sinopharm Chemical Reagent Co., Ltd. (Shanghai, China), Polyvinyl pyrrolidone (PVP, K29-32) and aniline (C_6_H_7_N, 99.5%) were purchased from Aladdin Industrial, and aniline was purified by vacuum distillation. Silkworm cocoons were provided by Shanghai Natural Wild-insect Kingdom Co. Ltd. (Shanghai, China). Dry air was purchased from Jiangsu Thorpe Group Co. Ltd. (Zhenjiang, China). The morphology characteristic of composite nanofibers were observed by scanning electron microscope (SEM, JSM-7800F, JEOL, Tokyo, Japan). Raman spectra were collected by XploRA Plus, HORIBA. Ltd. (Longjumeau, France).

### 2.2. Preparation of PANI/AgNWs Sensor on SFFs Substrate

#### 2.2.1. Preparation of AgNWs

In a typical synthesis [28], PVP and NaCl were dispersed in EG solution and then heated at 80 °C in a beaker for 20 min until dissolved. After that, PVP and NaCl solutions were mixed into a 100 mL conical flask. In the meantime, AgNO_3_ was dissolved in the mixed solution with stirring at a rate of 1000 rpm by a magnetic stirrer during the entire process. After stirring, the solution was evenly transferred to Teflon-sealed autoclaves. The Teflon-sealed autoclaves were heated in an electric thermostatic drying oven at 150 °C for 1 h and then cooled to room temperature. After that, the obtained solution was washed with deionized water three times. Finally, the concentration of purified AgNWs was 8.3 mg/mL measured by the density method.

#### 2.2.2. Preparation of SFFs

In order to remove sericin proteins, the silkworm cocoon was degummed two times with 0.05% Na_2_CO_3_ solution at 100 °C for 30 min obtaining SFFs [29]. Then, SFFs were washed with distilled water, filtrated under reduced pressure and dried at 40 °C in a draught-drying cabinet. The dried SFFs were ready for applications in research.

#### 2.2.3. Preparation of PANI/AgNWs/Silk Composites

PANI/AgNWs/silk composites were synthesized by in situ polymerization [30]. SFFs were first soaked in 0.5 mol/L H_2_SO_4_ containing AgNWs (0.5 mL, 1.0 mL, 1.5 mL and 3.0 mL) with magnetic stirring for 5 min to facilitate the adsorption of AgNWs on SFFs. Then, 0.04 mol/L aniline dissolved in 0.5 mol/L H_2_SO_4_ was added, stirring for 20 min. The 50 mL of 0.04 mol/L APS dissolved in 0.5 mol/L H_2_SO_4_ was added to the solution dropwise stirring for 2 h, 4 h, 6 h, 8 h. The solution was stirred in an ice-water bath at 0 °C for the entire process.

### 2.3. Pork and Pork Freshness Evaluation

#### 2.3.1. Pork Samples

Samples taken from longissimus muscles were collected from 21 pig carcasses in a local meat company (the Yurun Meat Company, Jiangsu, China) and transported to the laboratory in an incubator at the temperature of 4 °C. Upon arrival at the laboratory, the pork was cut into cubes with 2 cm × 2 cm × 1 cm after the fat and connective tissues had been removed in aseptic conditions. The pork samples were reweighed about 20 g, labelled and stored in a 4 °C refrigerator before analysis. Daily, six of prepared samples were randomly allocated to one of two processing treatments: measurement of TVB-N contents or test of PANI/AgNWs sensor. The entire survey was conducted over a 30-day period.

#### 2.3.2. Determination of TVB-N

Stream distillation method was used as the measuring method for TVB-N contents [31]. First, all samples were finely chopped individually using a homogenizer (FJ200, Biaomo, Changzhou, China). Mincemeat measuring 20 ± 0.001 g was mixed together with 100 mL deionized water, poured into a beaker flask and stood for 30 min at room temperature. Next, the filtered solution was mixed with 10 g/L MgO. The distilled solution was collected into 20 g/L boric acid and titrated with 0.01 mol/L HCl. TVB-N contents were calculated according to the following Equation (1):(1)$$\mathrm{TVB-N}\left(\mathrm{mg}\;/100\;g\right)=\frac{\left(V_{1}{-V}_{2}\right)\times c\times 14}{m\times \frac{10}{100}}\times 100$$
where *V*1 denotes the titration volume of the tested sample (mL), *V*2 denotes the titration volume of blank sample (mL), *c* denotes the concentration of HCl (mol/L) and *m* denotes the weight of the minced pork sample (g).

### 2.4. System for Testing Sensor

#### 2.4.1. Design and Fabrication of Testing System

The testing system consisted of a computer, a multi-meter, a gas chamber, a desiccator, two gas cylinders and a sensor (as shown in Figure 1). The entire testing part, including the multi-meter and the sensor, was controlled by the computer. The computer issued instructions to the multi-meter and sensor to collect test data. The multi-meter was mainly used to gather the real-time resistances when samples were tested. In particular, the sensor was used for acquiring gas signals. It was composed of PANI/Ag/silk composites and flexible flat cable (FFC, 0.5 mm interval). PANI/Ag/silk composites were stuck in an FFC by silver-epoxy adhesives (as shown in Figure 1).

#### 2.4.2. TMA Testing

Figure 1 showed the dynamic responses of the sensor to TMA with a homemade test device. The sensor was placed into a gas chamber (1500 mL in volume) and used for testing resistances reflecting the concentration of TMA. Before testing, the chamber air needed to be changed with dry air for the final TMA test. The TMA gas flowed in the gas chamber from the cylinder at a rate of 2 L/min. The gas flow and direction were controlled by relief value. The resistance of the sensor was measured by a multi-meter (Agilent 34410A, Palo Alto, USA) at 2 s intervals. The resistances of the tested nanofibers were measured continuously in air (*R*_0_) and trimethylamine (*R_i_*) to evaluate response values (*S*), as Equation (2) showed.
(2)$$S=\frac{R_{i}{-R}_{o}}{R_{o}}$$

#### 2.4.3. Pork Freshness Testing

Dynamic responses of the pork were investigated using the above system. First, the dry air went through the RV-1 and flowmeter into the gas chamber at a speed at 2 L/min. About 10 min later, the sensor was in a uniform environment and started to test the resistance of air. Next, the pork chamber was opened until the gas chamber was full of volatile, organic gases. Then, the resistance of the pork sample was tested after the volatile organic gases stayed steady. Finally, the chamber was injected with dry air again, and the resistance was continually tested. All of the above were a cycle for pork freshness testing by sensing system, and all operation steps were performed at room temperature.

### 2.5. Statistical Analysis

The sensitivity of the AgNW sensor exposed to TMA was evaluated by a factorial analysis of variance using the polynomial-fit model with the program Origin 2021. Since the system included three-time replicates, it needed to be included in the model as a random term. The polynomial fit model was fitted by least-square means by *t*-test (*p* < 0.05) [32].

## 3. Results and Discussion

### 3.1. The Synthesis Principle of PANI/AgNWs/Silk Composites

The synthesis principle of PANI/AgNWs/silk composites was illustrated in Figure 2. The tyrosine exposed to silk fibroin is considered as the redox active sites between the silk fibroin and silver ions at room temperature [33]. The Ag^+^ existed in AgNW solutions as adsorbed ions, so the compound effect of PANI and AgNWs can be achieved by static electricity adsorption and ion adsorption. The complexation between Ag^+^ with hydroxyl groups of tyrosine residue occurred on the SFFs backbone [34], which played a role as the reactive center to conjugate Ag^+^ and chains [35]. All of the above proved the silk to be a suitable substrate supporting the adhesion of PANI/AgNWs. The PANI/AgNWs and PANI/silk were intertwined and interlaced with each other forming a reticular structure, which strongly enhanced the electrical conductivity of the silk and provided more effective contact areas for gas.

### 3.2. Optimization of PANI/AgNWs/Silk Sensor

To select excellent composites for TMA sensors, the amount of AgNWs and the polymerization time should be optimized. First, the sensor was optimized for detection TMA with different amounts of AgNWs (0.5 mL, 1.0 mL, 1.5 mL and 3.0 mL). Figure 3a–d showed the typical response-recovery curves under 100 μg/L concentrations of TMA at room temperature based on the PANI/AgNWs/silk sensor. When composites were exposed to TMA, TMAs were absorbed on the surface of composite materials, which resulted in the resistance of the sensor increasing observably. Then, the resistance of the sensor began to reach saturation levels after 150 s. When the chamber was pumped in the dry air, the resistance of the sensor started to decrease. It was obvious that response values went up when the contents of AgNWs increased before 1.5 mL. It indicated that the sensitivity of the composites showed a significant improvement with the increase in AgNWs; however, when the AgNWs rose to 3.0 mL, the response value dropped sharply. The principal reason was that the addition of AgNWs contributed to forming the reticular structure. When composites are exposed in TMA, the TMA can contact with composites fully for the reticular structure. The special structure led to a higher electrical conductivity and response value. While over AgNWs formed an excess of cross nanowires destroying reticular structures, the response value went down sharply.

The performance of the PANI/AgNWs/silk sensor fabricated by polymerization times of 2 h, 4 h, 6 h and 8 h was shown in Figure 3e–h. The response value of composites polymerized for 4 h was much higher than that of composites polymerized for other times. The short reaction time of PANI combining with AgNWs caused a mutilated reticular structure. The bare PANI had a poor electrical conductivity leading to a low response value. While a long reaction time of PANI combined with AgNWs contributed to excessive PANI coating, this hindered trimethylamine from adsorbing composites and caused the response value to decrease as well. Therefore, 1.5 mL AgNWs and PANI polymerized for 4 h received the best response value when used for composites in the repeatability and stability test.

### 3.3. Characterization of the PANI/AgNWs/Silk Sensor

The research of energy dispersive spectroscopy was designed to further prove elements in PANI/AgNWs/silk composites. As shown in Figure 4a, the main elements, including carbon and oxygen, presented a relatively high peak, and they were the main compositions of silk. The characteristic peak of Ag element was also presented in the energy dispersive of spectra, which indicated the success of PANI/AgNWs/silk composites.

Figure 4b displayed the Raman spectra of the different composites. The spectra of PANI/AgNWs/silk and PANI/silk composites exhibited a peak at 1490 cm^−1^ (stretching vibrations of C=N belonging to quinonoid units) and a band at 1585 cm^−1^ (stretching vibration of C=C in the quinonoid ring) [36]. Compared with PANI, the peaks for stretching vibrations of C=C and C=N belonging to quinonoid units moved to higher wavenumbers in PANI/silk and PANI/AgNWs/silk. The phase shift of spectra indicated the interaction of polymers with peptide bonds existing in SFFs macromolecule. The peak at 1166 cm^−1^ belongs to bending vibrations of C−N in semi-quinonoid rings [37]. On the other hand, the band of PANI/AgNWs/silk had a shoulder peak at 1159 cm^−1^ for bending vibrations belonging to benzenoid rings [38]. Typical characteristic peaks of polyaniline were exhibited at 409 cm^−1^ and 411 cm^−1^ [39]. All of the above demonstrated that PANI/AgNWs/silk composites had been successfully synthesized.

### 3.4. Sensing Performance of the Sensor for TMA

It is primarily important to increase the selectivity for the target gas. As shown in Figure 5a, the response values of the PANI/AgNWs/silk sensor towards 100 μg/L trimethylamine, ammonia (NH_3_), hydrogen sulfide (H_2_S), water (H_2_O) and ethanol (C_2_H_6_O) were measured according to Section 2.4.2. Every sample represented the average response values in triplicate, and error bars were estimated from three replicate tests. It was obviously found that the response towards trimethylamine is higher than that of other gases, which indicated that the PANI/AgNWs/silk sensor had a high selectivity for trimethylamine detection. The PANI/AgNWs/silk sensor is sensitive to nitrogen functional groups, so the PANI/AgNWs/silk sensor had a response to trimethylamine and ammonia. As shown in Figure 5b, the response of sensor was monitored towards 100 μg/L TMA for five cycles. The response and recovery times were estimated to be about 90 s and 110 s, respectively. It was observed that the resistance response of the sensor was retained after repeated cycles.

Figure 6 showed the stability of the sensor for TMA tests for 30 days. The sensor showed good stability in 21 days with response values from 1.1906 to 1.0189. When the test went to the 30th day, the response value reached 68.35% compared to the first day. While the test went to the 21st day, the response value reached 85.58%.

Figure 7a showed the response–recovery curves of the PANI/AgNWs/silk sensor to different concentrations of TMA ranging from 3.33 μg/L to 1200 μg/L. The sensor exhibited rapid response and recovery to TMA in different concentrations and was sensitive to TMA as small as 3.33 μg/L. Figure 7b showed the response values of the sensor as functions of TMA concentrations. The regression equation was *y* = 0.2273 + 0.0065*x* − 2.33711 × 10^−6^*x*^2^ (*R*^2^ = 0.9900) for the PANI/AgNWs/silk sensor (*p* < 0.05). The TVB-N content and its 95% confidence band were calculated in Figure 7b, which meant that the confidence band covered approximately 95% of the real effect. Compared with the informed research, the sensor showed the faster response, higher sensitivity and speedy recovery.

### 3.5. Application in Pork Freshness Detection

TVB-N, the nitrogen-containing substances produced by protein decomposing with the presence of enzymes and bacteria in the process of pork spoilage, plays an important role in the detection of meat freshness [40]. The fabricated sensor was used to detect the freshness of pork. As a contrast measurement, the TVB-N content of pork was measured at the same time.

The pork with the TVB-N content of less than 15 mg/100 g was judged as fresh meat, according to GB 2707-2016 (China National Standard). Figure 8a showed that the response values of the fabricated sensor changed with TVB-N contents in seven days. The TVB-N contents had little change and were below 15 mg/100 g; therefore, all pork meats were classified as fresh meat in the first three days. Then, a rapid increase was found upon the TVB-N and response values with prolonging the storage time of the pork. The TVB-N contents increased from 17 mg/100 g (Day 4) to 63 mg/100 g (Day 7), leading to the pork being classified as decayed meat. The dynamic curve of the response value and TVB-N contents showed a similar tendency. Figure 8b showed the correlation between TVB-N contents to sensor response values with a significant difference (*p* < 0.05). More specifically, the equation of linear regression was determined as *y* = 106.6057*x* − 1.3891 (*R*^2^ = 0.9747), and the 95% confidence band indicated the reliability of test results. In Figure 8c, the residuals were randomly and evenly distributed on the sides of *X*-axis. Figure 8d showed a normal probability plot of the residuals, which indicated that the variance is normally distributed as well. There are no heteroscedasticity and inadequate fitting problems. It is considered that the fitting functions fully meet the use conditions. The PANI/AgNWs/silk sensor was successfully applied in the determination of pork freshness.

## 4. Conclusions

In conclusion, a novel gas sensor based on PANI/AgNWs/silk was prepared successfully for detecting pork freshness. The silk, as a natural organic backbone, provided additional active sites for the sensor. The PANI/AgNWs exhibited excellent electrical conductivity and were assembled to SFFs forming the gas sensor. The gas sensor was the optimal candidate for TMA detection with good repeatability, stability and high sensitivity, for which its lowest detection limit was 3.3 μg/L. The regression equation was *y* = 0.2273 + 0.0065*x* − 2.33711 × 10^−6^*x*^2^ (*R*^2^ = 0.9900) for the PANI/AgNWs/silk sensor (*p* < 0.05). The pork was classified as fresh meat in the first three days when TVB-N contents were below 15 mg/100 g; therefore, the sensor can be applied for the detection of pork freshness and the results of the evaluation was the same as GB 2707-2016. The sensor exhibited good gas-sensing properties and great promise for applications in the pork industry.

## Figures and Tables

**Figure 1 foods-11-02372-f001:**
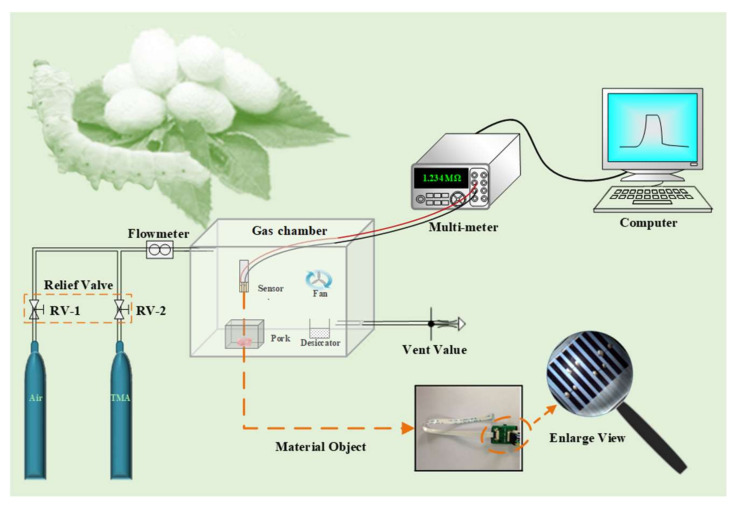
Schematic diagram of gas-sensing device and the photos of sensor.

**Figure 2 foods-11-02372-f002:**
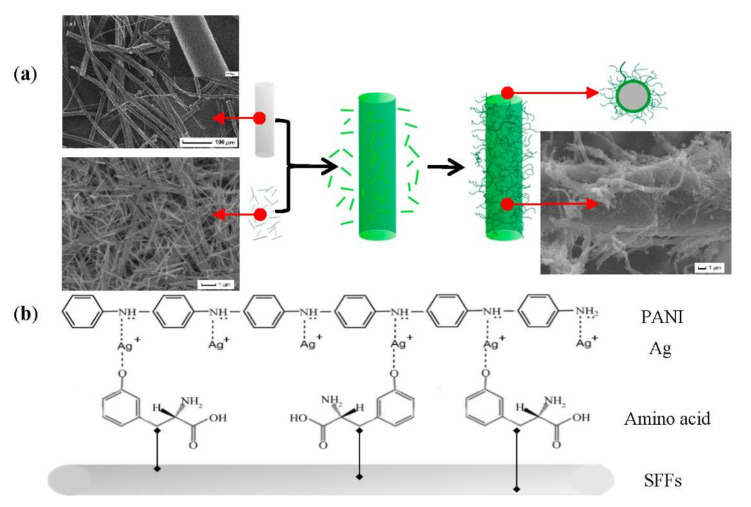
The formation process (**a**) and mechanism (**b**) of PANI/AgNWs/silk sensor. Insets in (**a**) show the enlarge SEM image of SFFs.

**Figure 3 foods-11-02372-f003:**
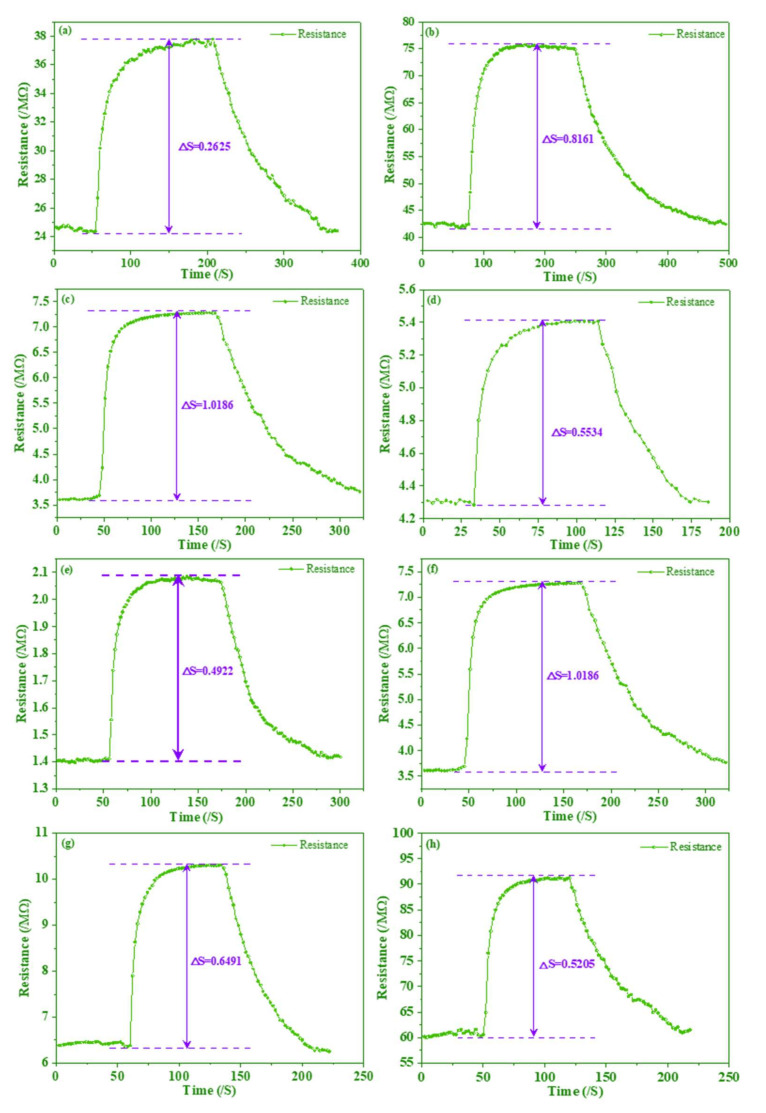
The response and recovery curves of the sensor towards 100 μg/L TMA with different volumes of AgNWs in (**a**–**d**) and different times in (**e**–**h**).

**Figure 4 foods-11-02372-f004:**
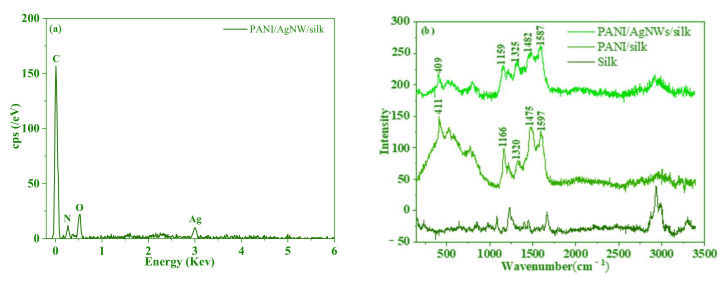
EDS spectrum of PANI/AgNWs/silk (**a**) and Raman spectra of the silk, PANI/silk and PANI/AgNWs/silk composites (**b**).

**Figure 5 foods-11-02372-f005:**
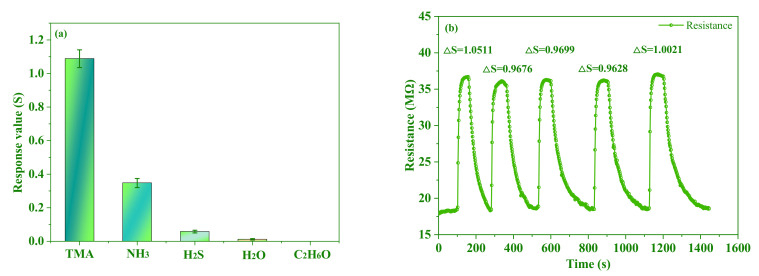
Sensor response of the PANI/AgNWs/silk composite nanofibers to 100 μg/L of different gases (**a**) and the response and recovery curves and the response value of the sensor towards 100 μg/L TMA for five cycles (**b**).

**Figure 6 foods-11-02372-f006:**
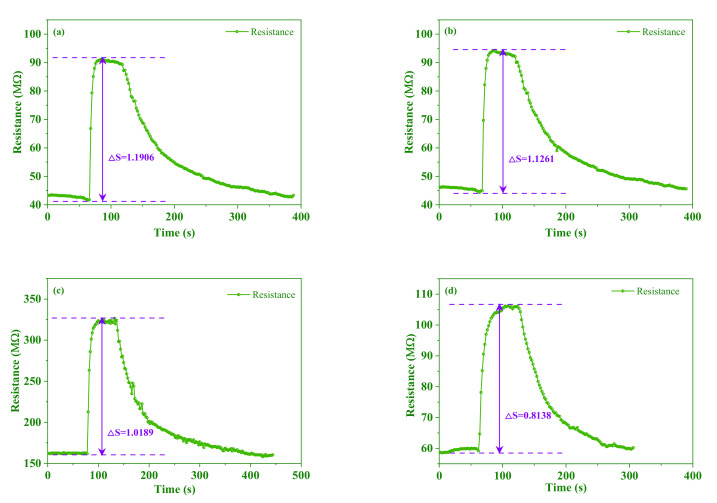
The sensing performance of PANI/AgNWs/silk sensors prepared with 1.5 mL AgNWs polymerizing 1.5 h to 100 μg/L TMA for one month in (**a**–**d**).

**Figure 7 foods-11-02372-f007:**
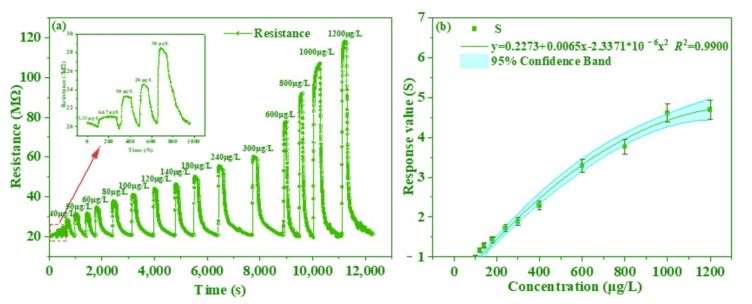
Resistance transients (**a**) and sensitivity (**b**) of the AgNWs sensor exposed to TMA of different concentrations at 25 °C.

**Figure 8 foods-11-02372-f008:**
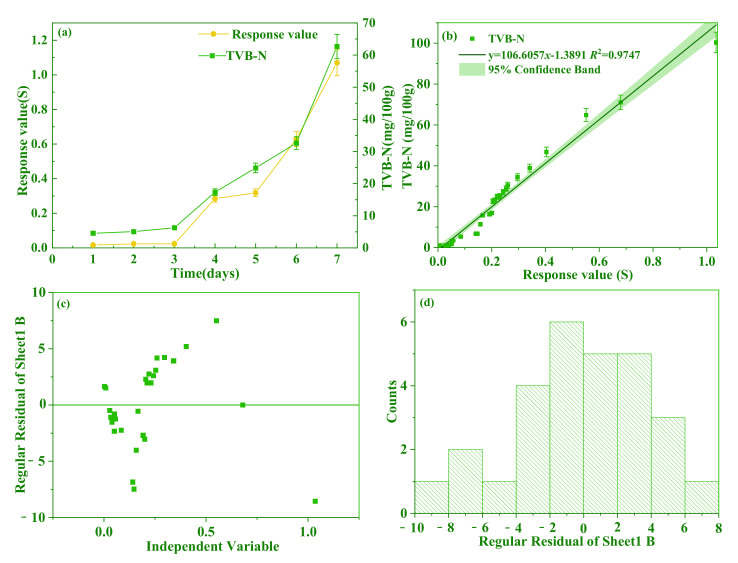
The changes of TVB-N and response value of the sensor to pork with storage time (**a**); the correlation between TVB-N contents to sensor response values (**b**); distribution diagram of regular residual (**c**); histogram of regular residuals (**d**).

## Data Availability

Data are contained within the article.

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
