# Peer review of "A Novel Gas Sensor for Detecting Pork Freshness Based on PANI/AgNWs/Silk"

_foods, 2022, doi:10.3390/foods11152372_

Round 1

Reviewer 1 Report

A room-temperature sensor for detecting the pork freshness based on PANI/AgNWs/silk

This research paper covers a relatively specific subject area and is therefore expected to be of general interest to readers. The paper describes a room-temperature sensor approach for detecting pork freshness based on PANI/AgNWs/silk

The following issues are presented for the authors' consideration in case the authors want to resubmit the paper:

·       The title is not appropriate. Revise it. It's not specific and reflects the content. “A room temperature”- but the abstract focuses focusing the gas sensor.

·       Write the abstract over again. The content arrangement is poor and needs to be improved. Make it more transparent and precise—lack of results, discussion, and application.

·       Introduction: Rewrite and rearrange the paragraph—lack of information related to the study's background. Include previous research, as well as the benefit of each material that has been used in the research study.

  What is the main question addressed by the research? 

-       How stable and specific is the gas sensor working at room temperature. 

-       What are the color changes if this paper focuses on the colorimetric?

-       How important is the silk in the sensor? What is the main role of silk?

Is it relevant and interesting? 

-       Yes, the results might be applicable as portable gas sensing.

How original is the topic? 

-       The topic covers the gas sensor for monitoring the meat freshness with the incorporation of silk. 

What does it add to the subject area compared with other published material? 

-       Application of silk. However, author should emphasize how important silk is in the proposed senor.

·      References are not appropriate. Pl ease use the updated references, especially in the introduction (example: 2018-2022). Read more journal papers. 

·       Line 33-40: Focus on the TMA? But the abstract focuses on TVBN.

·       Section 2.2.1: the author used the full name of “silver nanowires.” However, in section 2.2.2, “they used short form.” Make it synchronize. 

·       Line153 “slight amount,” please make it specific. “so the static electricity adsorption can achieve the compound effect of PANI and AgNWs and ion adsorption” described with citation.

·       Figure 2 SEM does not clear with the size.

·       Kindly submit to proofread.

·       Rewrite the conclusion, which does not reflect the manuscript's content.

· 

Author Response

Thank you for your suggestion. I have revised the manuscript.

Reviewer 2 Report

The article has innovation, however some adjustments will be necessary before its acceptance

Line 1 - What is the average consumption of pork meat? What are the benefits of consuming pork meat?

Line 2 - What factors make pork an easily perishable product? When using abbreviations, authors must define them before

What is the research hypothesis?

Insert the objective at the end of the introduction

Line 93 - Why did the authors choose the Longissimus muscle to carry out the research? What is the weight of the muscles? How many Longissimus muscles were used in the research? Were the Longissimus muscles obtained from the right and left sides of the carcasses? Was there a cleaning of the Longissimus muscles with the previous removal of fat and fascia before cutting the cubes? Were the dicing performed on the chilled samples or did they undergo a period at room temperature prior to dicing? How many cubes have been extracted from one muscle?

Line 97 - What is the storage period of the samples before carrying out the analyses? What is the average weight of the cubes? Daily... How many days did the survey last? How many samples were allocated per treatment over the duration of the research?

Line 149 - There are discussions in the Results item. Authors must separate the results of the discussions. There are two items: 3 Results; 4. Discussion.

The results are well presented, however lack of discussion. Explain why the answers obtained and justify them. The results should also be compared with previous studies on the topic addressed.

Line 272 – The conclusion shown is a continuation of the previous item. The conclusion should be brief and affirmative and answer the research hypothesis.

Author Response

(The authors gave the same response as above.)

Round 2

Reviewer 1 Report

The manuscript is well updated based on the suggestion, it just required minor changes in English proofreading.